# OpenReview forum: "Unlearned but Not Forgotten: Data Extraction after Exact Unlearning in LLM"
_NeurIPS.cc/2025/Conference — NeurIPS 2025 poster_

### Official Review · Reviewer_iY2B · 2025-06-19

**Clarity:** 3
**Significance:** 2
**Originality:** 2
**Rating:** 2
**Confidence:** 5

**Summary:**

This paper investigates privacy-oriented data extraction attacks against LLMs. The authors examine a scenario in which attackers can simultaneously access both pre-unlearning and post-unlearning model versions or their respective APIs. Their findings demonstrate that by combining the logit outputs from both model versions, attackers can enhance extraction rate beyond what would be possible using only the pre-unlearning model.

**Questions:**

**See Weakness.**

**Minor question:**

Does $q(x)$ in Sec.4.1 represent distribution of the gold standard “Retrain from Scratch” model?

**Ethical Concerns:**

["NO or VERY MINOR ethics concerns only"]

**Final Justification:**

I appreciate the authors’ effort in preparing the rebuttal and have understood the settings they considered. That said, I still remain concerned about the assumption that attackers have access to both the pre- and post-unlearning models. In many real-world scenarios, subsequent model versions often incorporate new training strategies or additional data corpora, which could lead to different dynamics. Furthermore, while the described setting is clear, it differs from the typical practical goal of unlearning—where model holders or red teams apply unlearning algorithms to remove harmful content before releasing the model—and, in the case of closed-source models, logits may not be accessible, further limiting applicability. Therefore, I tend to remain my original score.

**Limitations:**

Yes.

**Paper Formatting Concerns:**

No concern.

**Quality:**

2

**Strengths And Weaknesses:**

**Strengths:**

1.The paper follows a clear structure that makes it relatively easy to understand the authors' approach and findings.

2.Testing across three representative datasets. The authors have done a reasonably thorough job with their ablation studies.

**Weaknesses:**

I have a major concern over the paper's core assumption that attackers can access both pre-unlearning and post-unlearning models. This scenario largely defeats the purpose of unlearning methods, which exist precisely because the original models carry privacy risks we're trying to mitigate. Building an attack method that requires access to the pre-unlearning model isn't realistic in practice.

Even if we accept this setup, the main finding that combining outputs from both model versions improves extraction lacks sufficient novelty. The contribution is incremental over prior work. It's essentially like comparing a model that has memorized certain data with one that hasn't, then using the differences to reveal what was memorized. (Similar to previous results that leverage over-fitting to benefit privacy attacks). The information gain here shouldn't be surprising.

Further, I am not satisfied with the paper's writing logic, particularly with its bold "breaking the gold standard" title and challenging "retrain from scratch" approaches, and claims. Given the unrealistic attack scenario, the paper's claim of established unlearning practices doesn't really hold water.

---

> ### Author Rebuttal · Authors · 2025-07-29
>
> Thank you for your feedback!
>
> 1. W1: Attack assumption.
>
> > I have a major concern over the paper's core assumption that attackers can access both pre-unlearning and post-unlearning models. This scenario largely defeats the purpose of unlearning methods, which exist precisely because the original models carry privacy risks we're trying to mitigate. Building an attack method that requires access to the pre-unlearning model isn't realistic in practice.
>
> We focus on how unlearning is deployed in real-world scenarios, where our attack could realistically occur, especially for open-source models. It is entirely possible that an attacker has already pre-saved the pre-unlearning model—for instance, an adversary might successfully extract and release copyright-sensitive data (or the issue is otherwise discovered), prompting the model owner to perform unlearning. In such cases, our concern is that the attacker could then extract even more data than before unlearning was applied, creating a paradoxical situation where a mechanism intended to reduce leakage may actually amplify it.
>
> 2. W2: Contribution of Method.
>
> > The main finding that combining outputs from both model versions improves extraction lacks sufficient novelty. The contribution is incremental over prior work. It's essentially like comparing a model that has memorized certain data with one that hasn't, then using the differences to reveal what was memorized. (Similar to previous results that leverage over-fitting to benefit privacy attacks).
>
>
> This characterization oversimplifies our contribution. As detailed in Sec. 4, our method is not merely “comparing outputs.” We propose a principled **parametric approximation** that simulates the fine-tuning and unlearning dynamics to construct a model-guidance distribution, which is then used to drive extraction. This formulation is non-trivial and, to our knowledge, has not been explored in prior work.
>
> While the intuition that information gain can improve extraction may seem natural, our contribution lies in **how** we formalize and operationalize it. We demonstrate why this formulation works, and we show empirically that it consistently improves extraction performance. These insights are methodological, not just incremental.
>
> 3. Minor question:
>
> > Does  q(x) in Sec.4.1 represent distribution of the gold standard “Retrain from Scratch” model?
>
> No. As stated in Line 152, $q(x)$ represents the ground-truth probability distribution of the forgetting set $X_0$​, which we aim to approximate in order to enable strong extraction.

---

> > ### Comment · Area_Chair_Jsgi · 2025-08-04
> > **Reviewer interaction**
> >
> > I would kindly remind Reviewer iY2B to review the authors rebuttal and engage in the discussion period.
> > The authors have addressed all your raised points and are waiting for feedback.

---

> > ### Comment · Reviewer_iY2B · 2025-08-05
> >
> > Thank you for the detailed rebuttal! I have a follow-up question regarding the underlying assumptions of your evaluation setting. As I understand, your work assumes a grey-box threat model where an attacker has access to the output logits of both the pre-unlearning and post-unlearning models. This setup highlights that certain unlearning approaches may inadvertently increase the risk of exposing sensitive information. However, I wonder if this assumption aligns with the broader goals of unlearning research. Typically, one of the motivations behind designing unlearning algorithms is to ensure that no information from the pre-unlearning model is accessible after the unlearning process. If this assumption is relaxed, as in your setting, it seems that any unlearning method could risk additional leakage, simply due to increased exposure across two models (difference between two models are caused by unlearned materials). Could the authors elaborate further on this point? I’d greatly appreciate your clarification.

---

> ### Author Response · Authors · 2025-08-05
>
> Thank you for the thoughtful follow-up. We believe this discussion reflects a **difference in perspective** between how unlearning is typically **framed in the literature** and how it might be **deployed in realistic scenarios**.
>
> From our standpoint, unlearning—particularly if used in large-scale deployments—would often serve as a **reactive mechanism**, applied after a model has leaked personal information, raised copyright concerns, or been subjected to targeted extraction attempts. In such cases, it is entirely possible that the pre-unlearning model has already been downloaded, cached, or queried by adversaries. For example, once a model is released and subsequently found to have been exploited for privacy or copyright-related content, unlearning may be **invoked to mitigate future misuse**—making it harder for downstream users or future adversaries to access the exposed content. However, for those who have already interacted with the pre-unlearning model, unlearning may introduce new model differences that inadvertently assist further extraction attempts. For example, an attacker may have pre-saved the model weights, and the changes introduced by unlearning could provide additional signals that help guide future attacks.
>
> This motivates our decision to analyze unlearning under a **distinct threat model**, where the attacker has access to the logits APIs of both the pre- and post-unlearning models. While this setting differs from conventional assumptions, we argue that it is highly relevant in practice—particularly for released, open-sourced, or previously accessed models. In such contexts, unlearning is **not performed in isolation from the model’s prior exposure**; the behavioral differences introduced by unlearning may remain observable to adversaries who interacted with the original model.
>
> Our goal is to understand whether such differences, when exposed to adversaries with prior access, can unintentionally **amplify leakage**—through model guidance based on the differences between the two models. This perspective does not challenge the effectiveness of unlearning under its standard assumptions, but rather highlights that **additional risks may emerge when those assumptions no longer hold in deployment**.
>
> To avoid misinterpretation, we have revised our title and abstract to better reflect our focus on **exposure-driven vulnerabilities**, rather than suggesting any intrinsic algorithmic flaws. We hope this clarification helps position our work as a **complementary viewpoint**—one that expands the conversation around the robustness and deployability of unlearning in real-world systems. We appreciate the opportunity to elaborate on this point.
>
> Revised title and abstract are listed below. The bold parts indicate the modified text.
>
> ---
>
> **Title**: **Rethinking Exact Unlearning under Exposure:** Extracting Forgotten Data under Exact Unlearning in Large Language Models
>
> **Abstract:** Large Language Models are typically trained on datasets collected from the web, which may inadvertently contain harmful or sensitive personal information. To address growing privacy concerns, unlearning methods have been proposed to remove the influence of specific data from trained models. Of these, exact unlearning---which retrains the model from scratch without the target data---is widely regarded the gold standard **for mitigating privacy risks in deployment**. In this paper, we **revisit this assumption in a practical deployment setting where both the pre- and post-unlearning logits API are exposed, such as in open-weight scenarios. Targeting this setting,** we introduce a novel data extraction attack that leverages signals from the pre-unlearning model to guide the post-unlearning model, uncovering patterns that reflect the removed data distribution. Combining model guidance with a token filtering strategy, our attack significantly improves extraction success rates---doubling performance in some cases---across common benchmarks such as MUSE, TOFU, and WMDP. Furthermore, we demonstrate our attack's effectiveness on a simulated medical diagnosis dataset to highlight real-world privacy risks associated with exact unlearning. In light of our findings, which suggest that unlearning may, in a contradictory way, increase the risk of privacy leakage **during real-world deployments**, we advocate for evaluation of unlearning methods to consider broader threat models that account not only for post-unlearning models but also for adversarial access to prior checkpoints.

---

> > ### Comment · Reviewer_iY2B · 2025-08-09
> >
> > Thank you to the authors for the further clarification. I have understood the setting they considered, and agree that in the case of an open-sourced model where a later version follows the same architecture but applies unlearning methods, their setting could indeed arise. That said, in many real-world scenarios, later versions often adopt new training strategies or additional data corpora, which may lead to different dynamics. Moreover, while the described setting is clear, it differs from the common practical goal of unlearning—where model holders or red teams update a model to remove harmful content prior to release—and in the case of closed-source models, logits may not be accessible, further limiting applicability. I nevertheless appreciate the authors’ efforts in clarifying these points.

---

### Official Review · Reviewer_rHwh · 2025-06-22

**Clarity:** 3
**Significance:** 3
**Originality:** 3
**Rating:** 5
**Confidence:** 2

**Summary:**

The paper investigates the limitations of exact unlearning in LLMs, which is commonly considered the most reliable method for removing private data by retraining the model without it. The authors challenge this assumption by introducing a new type of extraction attack that exploits access to both the pre- and post-unlearning versions of a model. Th method uses the pre-unlearning model to guide the post-unlearning model's outputs, revealing patterns that correspond to the data that was meant to be forgotten. By combining this model guidance with a token filtering strategy, the propose method improves data recovery success rates across standard benchmarks like MUSE, TOFU, and WMDP, as well as in a realistic medical data scenario. The results suggest that, under practical conditions (e.g. older checkpoints are available), exact unlearning can paradoxically lead to greater privacy risks. As a provided conclusion, unlearning methods must be evaluated under broader threat models that consider access to earlier model states, not just the final unlearned model.

**Questions:**

- Can you demonstrate similar extraction effectiveness on at least one very large model (e.g., LLaMA-3-70B or Mixtral)? Showing this would strengthen the generality of your claims and would positively influence my score.

- In commercial settings where only completions (not logits) are exposed, how often would an attacker be able to retain usable pre-unlearning outputs? If feasible, can you run your attack using saved completions rather than logits to assess its practicality in more restrictive environments?

- Since hyperparameters are selected empirically, please provide clearer guidance or theoretical intuition for choosing them.

Overall I tend to be positive for this paper, but I might be missing important references in my assessment.

**Ethical Concerns:**

["NO or VERY MINOR ethics concerns only"]

**Final Justification:**

All the key issues raised during the review phase were addressed in the rebuttal, including additional experiments that highlight the method's benefits on larger models as well. For these reasons, as anticipated in my initial review, I raise my score to full Accept.

**Limitations:**

Yes.

**Paper Formatting Concerns:**

None.

**Quality:**

3

**Strengths And Weaknesses:**

I enjoyed reading the paper. The language is fluent, the motivation is well laid out, the threat model and notation are clear, and key equations are accompanied by intuitive diagrams (Fig. 2) that convey how guidance shifts token probabilities. The experimental results (especially ablation) make it easy to trace where gains come from. Some weaknesses remain: the manuscript is occasionally repetitive, the medical-data experiment is buried in an appendix and depends on a synthetic generation pipeline that is only sketchily described, and certain hyper-parameter choices (e.g., guidance scale selection) appear empirical rather than principled, which may hinder straightforward replication.

In technical terms, the paper is solid in that it formalises a realistic two-checkpoint threat model, proposes an effective guidance-and-filter algorithm, and validates on three public unlearning benchmarks plus a synthetic medical dataset. The observed improvements seem dramatic (to within my limited knowledge of the topic), roughly doubling strict extraction accuracy on MUSE and TOFU with Phi-1.5.

At the same time, the empirical section leans on relatively small models (Llama-2-7 B and Phi-1.5) and synthetic or already-public datasets, so readers are left to extrapolate to frontier-scale LLMs and truly proprietary data. Moreover, the method's core assumption (attacker access to pre-unlearning weights or logits) while plausible for open-weight releases, is not universally realistic and largely defines the success of the attack, so the threat surface may be narrower than the headline suggests.

I particularly appreciated the paper's perspective on overturning a widespread belief that "exact unlearning is the gold standard", showing that, under a common practical scenario, retraining from scratch can actually augment leakage risk. This insight will likely influence how the research community think about compliance with the "right to be forgotten" legislation, and it is a novel insight to the best of my knowledge.

Still, the broader impact depends on how often adversaries can save old checkpoints; in many commercial API settings this is not possible, which softens the immediate policy implications. The defense analysis is also preliminary: adding noise or unrelated data reduces leakage only modestly and at steep utility cost, but the paper stops short of proposing a robust fix, so its practical payoff is mainly diagnostic.

From the methodological viewpoint, combining classifier-free guidance with contrastive token filtering to target unlearning gaps seems like a clever adaptation of ideas from diffusion models and open-ended decoding, and applying that combination specifically to the exact-unlearning setting appears novel. However, I might be missing important prior work due to my limited expertise in this specific area.

Overall, the paper makes a timely point with good evidence and clear exposition, but its dependence on a particular assumption about model access and its limited exploration of stronger defenses dminish the potential impact.

---

> ### Author Rebuttal · Authors · 2025-07-29
>
> Thank you for your insightful feedback!
>
> 1. Weakness&Q2: Practical Attack Scenarios.
>
> >  Moreover, the method's core assumption (attacker access to pre-unlearning weights or logits) while plausible for open-weight releases, is not universally realistic and largely defines the success of the attack, so the threat surface may be narrower than the headline suggests.
>
> > In commercial settings where only completions (not logits) are exposed, how often would an attacker be able to retain usable pre-unlearning outputs? If feasible, can you run your attack using saved completions rather than logits to assess its practicality in more restrictive environments?
>
> We consider open-source scenarios (such as those on Hugging Face) to be among the most practically significant. For example, an attacker targeting a fine-tuned LLM may have previously extracted partial sensitive data using the original model and saved both the model and the corresponding outputs. Once the model is later updated to forget specific data, the attacker can re-run extraction using our method.
>
> We acknowledge that in API-only settings, access to logits is not always available. In some cases, such as with GPT-4o, log probabilities can be retrieved via the logprobs parameter. However, other APIs may not expose this information at all, which makes our attack more difficult to apply in those environments. One possible workaround is to approximate logits by sampling multiple times per token, though this approach is computationally expensive. Extending our attack to settings where only completions are available—such as by relying solely on cached outputs rather than logits—is an important direction for our future work.
>
> 2. Weakness&Q1: Extraction on Larger Models.
>
> > At the same time, the empirical section leans on relatively small models (Llama-2-7 B and Phi-1.5) and synthetic or already-public datasets, so readers are left to extrapolate to frontier-scale LLMs and truly proprietary data.
>
> > Can you demonstrate similar extraction effectiveness on at least one very large model (e.g., LLaMA-3-70B or Mixtral)? Showing this would strengthen the generality of your claims and would positively influence my score.
>
> We add results on Mixtral-8x7B-Instruct-v0.1 by LoRA‑fine‑tuning on TOFU for two epochs with a cosine learning rate of 1e‑4 and a 10% forgetting set (all other settings followed our default setting in Sec. 5.1). For extraction, we apply a guidance scale of w=2. The improvement is consistent, as shown in the table below:
>
> |     Extraction Methods    | rougle-L(R)	 |    A-ESR0.9   	| A-ESR1.0  |
> |:-------------------------:|:-----------:|:----------:|:----------:|
> | Post-Unlearning Generation | 0.372      	 | 0.015     	| 0.013	 |
> | Pre-Unlearning Generation   |    0.465   	 | 0.040    	| 0.018	 |
> |            Ours           	|    0.537    	| 0.103     	| 0.055	 |
>
> 3. Weakness&Q3: Hyperparameters Choosing.
>
> > certain hyper-parameter choices (e.g., guidance scale selection) appear empirical rather than principled,
>
> > Since hyperparameters are selected empirically, please provide clearer guidance or theoretical intuition for choosing them.
>
> We agree that the guidance scale w is an important hyper-parameter, and its choice is grounded in our theoretical formulation rather than being purely empirical. As shown in Eq. (2), w=1/λ controls the strength of the guidance, which determines how much the generation is influenced by the score difference between the pre- and post-unlearning models. Intuitively, this reflects how much the pre-unlearning model still retains information about the forgotten data.
>
> From a theoretical perspective, as discussed in Sec. 4.1, Eq. (2) assumes a reverse process in which the post-unlearning model is fine-tuned back toward the pre-unlearning model using the forgotten data. Under this view, the pre-unlearning model’s distribution can be seen as a mixture of the post-unlearning distribution and the forgotten data distribution. The guidance scale w thus controls how strongly we extrapolate from the post-unlearning model toward the forgotten data distribution. A larger w implies a more aggressive shift back toward the original state, while a smaller w corresponds to a gentler correction—aligning smoothly with the amount of residual memorization.
>
> Importantly, even without tuning w to its optimal value, our method consistently improves extraction performance over the baseline as long as w is not set too large. Specifically, applying model guidance to steer generation away from the post-unlearning distribution is always beneficial. As shown in Fig. 5 and Fig. 6, settings with w<2.0 consistently lead to better extraction results across both Phi and LLaMA-2-7B, compared to the baseline case of w=1.0.

---

> > ### Comment · Reviewer_rHwh · 2025-08-01
> >
> > Thank you for your clarifications -- especially the additional results! I'll raise my score.

---

### Official Review · Reviewer_MFiG · 2025-06-30

**Clarity:** 4
**Significance:** 3
**Originality:** 2
**Rating:** 4
**Confidence:** 4

**Summary:**

The paper presents a new extraction attack against unlearning, that can even be effective when retraining from scratch, which is known to be the gold unlearning approach. The key assumption is that the attacker has access to the model before and after unlearning. The attacker then exploits the log probability between the two models  to guide the generation. They also apply a contrastive decoding to constraint the candidate next tokens only to those with high probabilities under the pre-unlearning model. The results on different models and unlearning datasets show extraction improvement compared to naively extracting from pre- or post- unlearning models.

**Questions:**

please see my questions in the weaknesses section.

**Ethical Concerns:**

["NO or VERY MINOR ethics concerns only"]

**Final Justification:**

While the conceptual novelty of the proposed attack is limited, the paper presents the first reported results in this problem setting. The experimental evaluation is diverse and thorough. Additional results during the rebuttal resolved some of my concerns. With claims moderated and limitations explicitly stated, the work merits acceptance.

**Limitations:**

yes

**Quality:**

3

**Strengths And Weaknesses:**

**Strength**

**S1**: The paper studies an important threat to exact unlearning that is the first time explored for extraction from LLMs.

**S2**: The paper is very well written and presented.

**S3**: Having access to both pre- and post- unlearning models is a practical assumption especially in today’s open source LLM status.


**Weaknesses**


**W1**: All the core ideas of the paper root in recent papers, reducing the novelty. For example, using model differences before and after unlearning was first proposed by Chen et al 2021, to devise a new membership inference attack. This paper doesn’t cite that work.

**W2**: I feel like the key message in the title and abstract is a bit overstated. As currently stated, it suggests that retraining-from-scratch itself is vulnerable, which could be misinterpreted as a failure of the algorithmic guarantee. In reality, the threat arises from the system-level exposure of both pre- and post-unlearning models—rather than a shortcoming of the unlearning method per se. It is like saying that DP is not private if there is a third signal that leaks the difference between the neighbouring datasets.


**W3**: In the defence methods, where noise is added to simulate DP-SGD, an important aspect of dp-sgd that is gradient clipping is neglected. Do you have any insights about the effect of gradient clipping?

**W4**: The scenario taken by this paper is to fine-tune the entire model parameters on the unlearning datasets (tofu etc.). But in practice, Peft finetuning is very popular. Do the authors have any insights how this will change the attacks success rate?

**W5**: In your setup, original model and the gold model have the same initialization due to the pretrained models. Do you think if there was a way to change the initializations, the attacks success rate would change?


**W6**: Is A-ESR calculated over the entire sample length? Is there a way to factor out the effect of the known prefix?



Chen, Min, et al. "When machine unlearning jeopardizes privacy." Proceedings of the 2021 ACM SIGSAC conference on computer and communications security. 2021.

---

> ### Author Rebuttal · Authors · 2025-07-29
>
> Thank you for your insightful feedback!
>
> 1. W1: Missing Citation for related work.
>
> > The core ideas of the paper root in recent papers, reducing the novelty. For example, using model differences before and after unlearning was first proposed by Chen et al 2021, to devise a new membership inference attack. This paper doesn’t cite that work.
>
> Thank you for pointing this out. We acknowledge that we missed citing this highly relevant work, which also considers privacy risks by comparing models before and after unlearning. We will include the citation in the revised version.
>
> Our main methodological contribution lies in making model differences effective for data extraction in the context of large language models. Since the learned manifolds of LLMs are highly structured and aligned with natural language, we operate within these manifolds using guidance-based generation. This contrasts with optimization-based methods, which might produce unnatural or incoherent outputs.
>
> 2. W2: Overstated title:
>
> >  In reality, the threat arises from the system-level exposure of both pre- and post-unlearning models—rather than a shortcoming of the unlearning method. It is like saying that DP is not private if there is a third signal that leaks the difference between the neighbouring datasets.
>
> We agree that the threat primarily stems from the exposure of both the pre- and post-unlearning models, rather than from a fundamental flaw in the unlearning algorithm itself. Our main takeaway is to caution against applying exact unlearning in cases where the pre-unlearning model may already have been exposed—such as in open-source scenarios where earlier versions could have been downloaded and saved. In such cases, exact unlearning might inadvertently increase risk rather than mitigate it. Accordingly, we will revise the title and framing to emphasize the importance of careful consideration during the deployment of unlearning, rather than suggesting a failure of the algorithmic guarantee itself.
>
> 3. W3: Gradient clipping in DP-SGD:
>
> > In the defence methods, where noise is added to simulate DP-SGD, an important aspect of dp-sgd that is gradient clipping is neglected. Do you have any insights about the effect of gradient clipping?
>
>
> Thanks for pointing this out. In Noisy Gradient Updates (Sec 5.6), we treat the absolute noise scale as our primary DP-SGD variable and fix the clipping norm at 2. **Empirically**, we observe that, under our mini‑batch training regime, typical choices of the clipping norm have virtually no measurable impact on final model performance. We further conducted experiments with clipping norms ranging from 0.1, 0.25, 0.5, 1, 2, to 4, finding that neither utility nor extraction rate changes more than 1%.
>
> **Theoretically**,  we define the batch noise scale $\eta = \frac{clipping\\_norm \times noise\\_multiplier }{batch\\_size}$. Equivalently (see Sec. 5.6), if $\alpha$ denotes the per‑example noise scale, then $\eta = \frac{\sigma}{batch\\_size}$. In our experiments, with batch size of 4, this noise scale $\eta$ far outweighs the typical gradient magnitudes, so the injected noise dominates the update. In particular, let $N$ denote the number of fine‑tuning parameters, since:
> $$
> \mathbb{E}\|gradient\\_noised\|^2 \approx \|gradient\\_clipped\|^2 +  \mathbb{E}\|noise\|^2 = \|gradient\\_clipped\|^2 + N \eta^2
> $$
> For Phi-1.5 (fintuning parameters $N > 5\times10^{5}$), even with $\eta=0.01$, the expected gradient norm after noise addition exceeds $50$, while original gradients are clipped to $\approx1$. The noise effect overwhelms the clipping constraint, making clipping variations irrelevant to defense effectiveness.
>
> This is why we focus solely on noise scales as the key defense factor in our paper.
>
>
> 4. W4: Peft finetuning:
>
> >  But in practice, Peft finetuning is very popular. Do the authors have any insights how this will change the attacks success rate?
>
> We add results on Phi-1.5 by LoRA‑fine‑tuning on TOFU for five epochs with a constant learning rate of 2e‑4 and a 10% forgetting set (all other settings followed our default setting in Sec. 5.1). Then, we employ a guidance scale w=1.6 for our extraction, increasing the extraction rate. The improvements are consistent, as shown in the following table.
>
> |     Extraction Methods    | rougle-L(R)	 |    A-ESR0.9   	| A-ESR1.0  |
> |:-------------------------:|:-----------:|:----------:|:----------:|
> | Post-Unlearning Generation | 0.412      	 | 0.010     	| 0.003	 |
> | Pre-Unlearning Generation   |    0.544   	 | 0.070    	| 0.055	 |
> |            Ours           	|    0.614    	| 0.160     	| 0.108	 |
>
>
> 5. W5: Different initializations of the models:
>
> > In your setup, original model and the gold model have the same initialization due to the pretrained models. Do you think if there was a way to change the initializations, the attacks success rate would change?
>
> We agree that in our setup, both models share the same initialization due to being derived from the same pretrained model. In practice, it may be difficult to alter the initialization meaningfully. However, if a "twin" model exists—i.e., a model trained with different random seeds or slightly different settings but that still learns a similar distribution—the unlearning could be applied to this unreleased twin version. This might reduce the attack's effectiveness to some extent. However, since our approach is based on the learned data distribution rather than the exact model weights, the attack should still be partially effective as long as the two twin models learn sufficiently similar distributions.
>
> 6. W6: About metric A-ESR.
>
> >  Is A-ESR calculated over the entire sample length? Is there a way to factor out the effect of the known prefix?
>
>
> No, A-ESR is computed only over the generated suffix, excluding the known prefix. Therefore, the effect of the known prefix is already factored out by design.

---

> > ### Comment · Reviewer_MFiG · 2025-08-02
> >
> > Thanks for the clarifications and the extra results. curious to hear the new framing of the title and the abstract.

---

> ### Author Response · Authors · 2025-08-03
> **Revised Title and Abstract**
>
> Thank you for your comments! We plan to tone down the title and revise the abstract as follows to better highlight the risks of unlearning under exposure in practical deployments, rather than the flaws of the algorithm itself. The bold parts indicate the modified text.
>
> **Title**: **Rethinking Exact Unlearning under Exposure:** Extracting Forgotten Data under Exact Unlearning in Large Language Models
>
> **Abstract:** Large Language Models are typically trained on datasets collected from the web, which may inadvertently contain harmful or sensitive personal information. To address growing privacy concerns, unlearning methods have been proposed to remove the influence of specific data from trained models. Of these, exact unlearning---which retrains the model from scratch without the target data---is widely regarded the gold standard **for mitigating privacy risks in deployment**. In this paper, we **revisit this assumption in a practical deployment setting where both the pre- and post-unlearning logits API are exposed, such as in open-weight scenarios. Targeting this setting,** we introduce a novel data extraction attack that leverages signals from the pre-unlearning model to guide the post-unlearning model, uncovering patterns that reflect the removed data distribution. Combining model guidance with a token filtering strategy, our attack significantly improves extraction success rates---doubling performance in some cases---across common benchmarks such as MUSE, TOFU, and WMDP. Furthermore, we demonstrate our attack's effectiveness on a simulated medical diagnosis dataset to highlight real-world privacy risks associated with exact unlearning. In light of our findings, which suggest that unlearning may, in a contradictory way, increase the risk of privacy leakage **during real-world deployments**, we advocate for evaluation of unlearning methods to consider broader threat models that account not only for post-unlearning models but also for adversarial access to prior checkpoints.

---

### Official Review · Reviewer_pLAx · 2025-07-02

**Clarity:** 3
**Significance:** 3
**Originality:** 3
**Rating:** 5
**Confidence:** 4

**Summary:**

This paper challenges the traditional exact-unlearning paradigm and demonstrates that adversaries can effectively extract the very date to be forgetten with the query access to both pre-unlearned and post-unlearned models.
To exploit this vulnerability, the authors leverages the pre-unlearned model as the reference model and draw inspirations from contrastive decoding techniques, guiding the model generations towards the forgett
The empirical evaluations includes several standard unlearning benchmarks (e.g. MUSE, TOFU, WMDP) and a self-designed dataset using Llama-2-7b and Phi-1.5.
Experimental results consistently demonstrate the effectiveness of proposed method ``model guidance", largely outperforms baseline approaches.

**Questions:**

Q1. The considered threat model that adversaries can obtain access to both pre-unlearned and post-unlearned models is relatively novel. I wonder whether this setting can work under realistic scenarios, and is there any references can be provided?
How will the overall framework run when adversaries can only get access to the post-unlearned model?

Q2. The statement that ``The behavioral divergence between the two models encodes rich information about the removed data." is interesting to me. Could you provide more discussions or explanations from both empirical and theoretical perspectives?

Q3. I notice that some references and citations about data extraction and exact unlearning are a little out ot date. I suggest the authors to update certain references in the next version.

Q4. I also care about the effectiveness of ``model guidance" under approximate unlearning settings, could you please provide more discussions?

Q5. To the best of my knowledge, data extraction attacks remain highly sensitive to parameter selection [2]. However, the proposed ``model guidance" introduces multiple issues about parameter tuning (e.g. guidance scale, filter strictness, and training iterations).
I am afraid this would bring difficulties under realistic applications.

[2] Bag of Tricks for Training Data Extraction from Language Models. ICML 2023.

**Ethical Concerns:**

["NO or VERY MINOR ethics concerns only"]

**Final Justification:**

The authors provide detailed responses that address all my concerns, so I unpdate my socre accordingly.

**Limitations:**

Yes.

**Paper Formatting Concerns:**

No formatting concerns.

**Quality:**

3

**Strengths And Weaknesses:**

Strengths
- This paper concentrates on a highly relevant and previously under explored scenario for challenging the traditional paradigm of exact unlearning, which is novel and significant.
- The proposed extraction strategy leverages the pre and post unlearned models for utilitzing the reference-based extraction methods and contrastive decoding techniques. The design is straightforward and intuitive.
- The authors evaluate their methods across three classic unlearning benchmarks and their new designed dataset, improving the robustness and convincement of experimental evaluations.
- The paper is well-written and easy to follow.

Weaknesses
- This paper highlights that when adversaries obtain access to both pre-unlearned and post-unlearned models, the data security of forgetten data is not well guaranteed. However, this assumption is a relatively strong setting and this might not be hold under all unlearning scenarios.
Thus, I might consider the statement that ``break the gold standard" to be somewhat overclaimed to me. In addition, this paper seems to lack detailed discussions about approximate unlearning, which stands as another commonly studied topic.
- Despite the significance of the concentrated topic, the proposed solution is not that surprising to me. The necessity and motivation of utilizing contrastive decoding should be provided.
- There are only two models to be tested. I actually understand that MUSE and TOFU only includes two models (Phi-1.5 and Llama-7-2b), so the authors only conduct evaluations with both models. However, since the authors introduce their own designed dataset, I will appreciate it if more models (e.g., Mistral or Qwen) can be tested so that we can tell whether model architectures have certain impact on the effectiveness of the proposed method.
- The tested baseline only includes the generations of pre-unlearned and post-unlearned models. I suggest the authors to consider more SOTA data extraction techniques (e.g. [1]).
- The proposed method introduces multiple issues of parameter tuning, which may hinder its application under realistic scenarios (See Question part).

[1] ETHICIST: Targeted Training Data Extraction Through Loss Smoothed Soft Prompting and Calibrated Confidence Estimation. ACL 2023.

---

> ### Author Rebuttal · Authors · 2025-07-29
>
> Thank you for your insightful feedback!
>
> 1. W1: Overclaim of the title
>
> > … However, this assumption is a relatively strong setting and this might not be hold under all unlearning scenarios. Thus, I might consider the statement that ``break the gold standard" to be somewhat overclaimed to me.
>
> We agree that our attack mostly applies to scenarios where both model checkpoints are available or where the APIs provide sufficient access. Accordingly, we will revise the claim from “break” to a softer framing, such as “rethink” or “re-evaluate.” Our focus on exact unlearning stems from its role as a reference point in current benchmarks—such as MUSE and TOFU—which treat it as the ideal case of what unlearning can achieve. Through this paper, we hope to draw attention to the fact that even such idealized unlearning may still expose significant privacy risks.
>
>
>
> 2. W1&Q4: lack discussions of approximate unlearning.
>
> > In addition, this paper seems to lack detailed discussions about approximate unlearning
>
> > I also care about the effectiveness of ``model guidance" under approximate unlearning settings, could you please provide more discussions?
>
> We include experiments on approximate unlearning methods in Appendix Sec. C, where our extraction method consistently outperforms the baseline. Notably, extraction becomes more effective when the approximate unlearning better preserves model utility. Since exact unlearning yields the best utility by retraining from scratch, the model guidance direction remains well aligned with the original data distribution, making it more effective. In contrast, approximate unlearning methods typically degrade utility to some extent, causing the guidance direction to partially deviate from the true data distribution.  In conclusion, our method consistently improves extraction performance when the post-unlearning model preserves a certain level of utility.
>
>
>
> 3. W2: Necessity of the method
>
> > Despite the significance of the concentrated topic, the proposed solution is not that surprising to me. The necessity and motivation of utilizing contrastive decoding should be provided.
>
>
> We view it as a feature—not a bug—that our attack method is relatively simple and natural. This demonstrates that even when an attacker is not very sophisticated, exact unlearning may still expose vulnerabilities.
>
> We are not sure that contrastive decoding is strictly necessary for performing the extraction attack. However, it is a natural strategy given that the primary goal of our paper is to extract part of the training data by effectively contrasting two model checkpoints—before and after unlearning. This contrast is derived from a parametric approximation (Sec. 4.1) that simulates the unlearning process and enables direct guidance in generation based on the difference between the two models.  Designing extraction attacks without relying on contrastive decoding would be a very interesting direction and is worth exploring in future work.
>
>
>
>
> 4. W3: Results on more models
>
> > I will appreciate it if more models (e.g., Mistral or Qwen) can be tested so that we can tell whether model architectures have certain impact on the effectiveness of the proposed method.
>
>
> We add results on **Mixtral-8x7B-Instruct-v0.1** by LoRA fine‑tuning on TOFU for two epochs with a cosine learning rate of 1e‑4 and a 10% forgetting set (all other settings followed our default setting in Sec. 5.1). For extraction, we apply a guidance scale of w=2. The improvement is consistent, as shown in the table below:
>
> |     Extraction Methods    | rougle-L(R)	 |    A-ESR0.9   	| A-ESR1.0  |
> |:-------------------------:|:-----------:|:----------:|:----------:|
> | Post-Unlearning Generation | 0.372      	 | 0.015     	| 0.013	 |
> | Pre-Unlearning Generation   |    0.465   	 | 0.040    	| 0.018	 |
> |            Ours           	|    0.537    	| 0.103     	| 0.055	 |
>
> 5. W4: More extraction Baseline:
>
> > The tested baseline only includes the generations of pre-unlearned and post-unlearned models. I suggest the authors to consider more SOTA data extraction techniques (e.g. [1]).
>
> Thank you for the suggestion and the reference. We agree that comparing against more SOTA extraction methods is valuable. Accordingly, we include additional experiments using ETHICIST on the TOFU dataset with Phi-1.5, under the 10% forgetting setting, as a comparison. Notably, ETHICIST assumes a stronger threat model that the attacker has access to part of the model’s training data. To align with this setting, we assume the attacker has access to half of the retained set in TOFU (while remaining blind to the target extraction set, i.e., the forgetting set), and use it to train the soft prompt. The hyper-parameters of ETHICIST follow the configuration provided in the original paper’s open-sourced code.
>
> Experimental results show that ETHICIST alone can partially improve the extraction rate. However, since ETHICIST is designed to extract training data from a single model, it is complementary rather than conflicting with our method, which leverages the difference between pre- and post-unlearning models. In fact, combining ETHICIST with our guidance-based approach leads to even stronger extraction performance.
>
> |     Extraction Methods    | rougle-L(R) |  A-ESR0.9 | A-ESR1.0  |
> |:-------------------------:|:-----------:|:---------:|-----------|
> | Pre-Unlearning Generation | 0.566       | 0.100     | 0.070     |
> |          ETHICIST         |    0.570    | 0.118     | 0.073    |
> |            Ours           |    0.643    | 0.202     | 0.120     |
> |      Ours + ETHICIST      |  **0.652**  | **0.213** | **0.153** |
>
>
>
> 6. W5 & Q5: Parameter tuning issues:
>
> >The proposed method introduces multiple issues of parameter tuning, which may hinder its application under realistic scenarios
>
> > the proposed ``model guidance" introduces multiple issues about parameter tuning (e.g. guidance scale, filter strictness, and training iterations). I am afraid this would bring difficulties under realistic applications.
>
> We agree that our method performs best under well-tuned hyper-parameters, and we acknowledge that such tuning may not always be feasible in real-world scenarios. However, we emphasize that our method consistently outperforms the baseline as long as the guidance weight w is not set too large. In particular, applying model guidance to steer generation away from the post-unlearning distribution is consistently beneficial. As shown in Fig. 5 and Fig. 6, settings with w<2.0 yield much better extraction results across both Phi and LLaMA-2-7B compared to the baseline case of w=1.0.
>
>
> 7. Q1: More discussion on the threat model:
>
> > I wonder whether this setting can work under realistic scenarios, and is there any references can be provided? How will the overall framework run when adversaries can only get access to the post-unlearned model?
>
> We consider open-weight scenarios (such as those on HuggingFace) to be the most practically significant. For example, an attacker targeting a medical fine-tuned LLM may have partially extracted sensitive data using the original model and saved both the model and the results. Once the model is later updated to forget specific data, the attacker can re-run extraction using our method. As for related references, while we are not aware of prior work specifically addressing this threat model in the LLM setting, similar targets on extracting unlearning data have been explored for simple models, including linear regression[1].
>
> When the attacker only has access to the post-unlearned model, our current method cannot be directly applied. However, it might be possible to adapt the framework by using a general-purpose language model as a reference to construct alternative guidance. This would require a different formulation and methodology, which we leave for future work.
>
> [1]  M. Bertran, S. Tang, M. Kearns, J. H. Morgenstern, A. Roth, and S. Z. Wu. Reconstruction attacks on machine unlearning: Simple models are vulnerable. Advances in Neural Information Processing Systems, 37:104995–105016, 2024.
>
>
> 8. Q2: More explanation on the method part:
>
> > The statement that ``The behavioral divergence between the two models encodes rich information about the removed data." is interesting to me. Could you provide more discussions or explanations from both empirical and theoretical perspectives?
>
> From an empirical perspective, we observe many cases like the one shown in Fig. 2, where the token most indicative of the forgotten data is not the one with the highest probability under either the pre- or post-unlearning model, but rather the one whose probability drops most significantly between the two. This suggests that the behavioral divergence between models reflects information specific to the removed data.
>
> From a theoretical perspective, the transition from Eq. (2) to Eq. (3) in Sec. 4.1 illustrates why such divergence encodes meaningful information. Specifically, in Eq. (2), we assume a reverse process in which the post-unlearning model could be fine-tuned back to the pre-unlearning model by training it again on the forgotten data for a fixed number of steps. Under this view, the pre-unlearning model’s distribution can be regarded as a mixture of the forgotten data distribution and the post-unlearning model’s distribution. This implies that the pre-unlearning model retains partial information about the forgotten data and requires guidance from the post-unlearning model to fully reconstruct it. The equivalence between Eq. (2) and Eq. (3) supports this interpretation.
>
>
> 9. Q3: Reference Updates
>
> > I notice that some references and citations about data extraction and exact unlearning are a little out ot date. I suggest the authors to update certain references in the next version.
>
> Thank you for the suggestion. We will incorporate more recent and relevant references in the next revision.

---

> > ### Comment · Reviewer_pLAx · 2025-08-05
> >
> > Thank you for your detailed response. I have no more concerns and have unpdated my socre accordingly. Please explicitly address the specific concerns raised in the review comments within the main text of the manuscript. This will ensure clarity for readers and provide necessary context for the modifications made.

---

### Official Review · Reviewer_SsS4 · 2025-07-02

**Clarity:** 4
**Significance:** 2
**Originality:** 3
**Rating:** 4
**Confidence:** 4

**Summary:**

The paper introduces a novel data extraction method for exact unlearned LLMs. Exact unlearned LLM refers to LLMs that were retrained without the extraction target text included in the training data. Exact unlearning was considered to be the safest “gold standard” option. However, the paper suggests that with their new data extraction attack, exact unlearning could actually backfire causing more security/privacy risks. The data extraction method assumes access to either the weights or the logits of both pre-unlearned LLM and exact unlearned LLM. Leveraging the exact unlearned LLM’s logits as reference, the attack then can select tokens that have high probabilities in the pre-unlearned LLM whilst having relatively low probabilities in the unlearned LLMs, which likely reflects the distribution of the extraction target text. The extraction attack proves to be effective on unlearning benchmarks with significant improvement in extraction metrics.

**Questions:**

- Can the authors please elaborate on how their method would work on chat models? I'm not really expecting for additional experimental results considering that will incur a lot of load, just sharing your thoughts will suffice.
- Were there any real cases where exact unlearning was used on an LLM?

**Ethical Concerns:**

["NO or VERY MINOR ethics concerns only"]

**Final Justification:**

The setting where the findings of this work is applicable is quite limited, which is a core limitation raised by other reviewers.

Overall, however, I do think that the findings of this work is interesting and unique, and in favor of acceptance.

**Limitations:**

Yes

**Quality:**

2

**Strengths And Weaknesses:**

**Strengths**

- The writing is solid with no ambiguities.
- The authors conduct a wide range of analysis and ablations which is appreciated.
- The paper is open about the limitations of its suggested approach, stating where it excels and where it does not.
- The paper highlights a potential risk of exact unlearning, which by previous works was considered to be secure and considered as “gold standard”.
- The paper contributes a new extraction method that seems to work very well.

**Weaknesses**
- From the results of Figure 5, the correct selection of w seems to matter significantly. The optimal value for w seems to depend on two factors: the backbone LLM and the training epoch. However, in a realistic scenario, the attacker will not have access to the information on the training epoch, nor have the access to the ground truth extraction data for tuning. Therefore, it's unclear how the attacker can guess a good value for w.
    - For the results in Table 1, the authors used the most optimal w for Phi and Llama (which I assume were tuned using the benchmark datasets) which shows the “best case” performance of the attack and might not be realistic.

- A practical setting where this attack can be applied is quite limited.
    - For the open-weight setting, the attacker has to have access to the weights of both the pre-unleared LLM and post-unlearned LLM, which is already a pretty strong assumption in my opinion.
    - For the API-only setting, wouldn’t the attacker need access to both the pre-unlearned and post-unlearned API during inference time, unlike how the authors claim in Line 116? The logits need to be recalculated for both pre- and post-unlearned LLMs every generation step. The chances of the API provider hosting both LLMs is highly unlikely. Also, proprietary LLMs rarely provide logits these days.
    - The efficacy of the attack was only demonstrated on base models. It’s unclear whether the attack also performs well on chat models, which is the predominant way of LLM deployment these days.
    - The attack assumes exact unlearning, which I’m not sure is commonly adopted in practice. A previous work points out that exact unlearning is “too costly to be practical” [1].

[1] MUSE: Machine Unlearning Six-Way Evaluation for Language Models, ICLR 2025

---

> ### Author Rebuttal · Authors · 2025-07-29
>
> Thank you for your insightful feedback!
>
> 1. About the hyper-parameter w:
>
> > From the results of Figure 5, the correct selection of w seems to matter significantly…it's unclear how the attacker can guess a good value for w.
>
> Our modeling begins with approximating how much the original model memorizes the dataset, as described in Eq. (2). This is described by the parameter λ, and since w=1/λ, the choice of w plays a central role in determining extraction performance. While the attacker may not have access to precise information about the extent of memorization, we emphasize two key points:
> First, our method consistently improves extraction performance over the baseline as long as w is not set too large. Specifically, applying model guidance to steer generation away from the post-unlearning distribution is always beneficial. As shown in Fig. 5 and Fig. 6, settings with w<2.0 consistently lead to better extraction results across both Phi and LLaMA-2-7B compared to the baseline case where w=1.0.
>
> Besides, in practice, partial side information about training iterations may still be available sometimes. For example, training epochs are sometimes disclosed in open-source models (e.g., open-r1/OpenR1-Qwen-7B on Huggingface). Additionally, prior work[1] suggests a correlation between model confidence and overfitting, which can serve to infer the training side information and could be potentially used to estimate the suitable guidance scale w.
>
> [1] Song, Liwei, and Prateek Mittal. "Systematic evaluation of privacy risks of machine learning models." 30th USENIX security symposium (USENIX security 21). 2021.
>
>
>
> 2. For the practical setting issues:
>
> > For the open-weight setting, the attacker has to have access to the weights of both the pre-unleared LLM and post-unlearned LLM, which is already a pretty strong assumption in my opinion.
>
> While the reviewers consider access to both the pre- and post-unlearning models a strong assumption, we argue that it is reasonable in open-weight scenarios. Open-weight models (such as those on HuggingFace) can be freely downloaded and archived before updates. For example, an attacker targeting a medical fine-tuned LLM may have partially extracted sensitive data using the original model and saved both the model and the results. Once the model is later updated to forget specific data, the attacker can re-run the extraction using our method.
>
> > For the API-only setting, wouldn’t the attacker need access to both the pre-unlearned and post-unlearned API during inference time, unlike how the authors claim in Line 116? The logits need to be recalculated for both pre- and post-unlearned LLMs every generation step…., proprietary LLMs rarely provide logits these days.
>
> In some API-only settings, log probabilities can still be accessed—for example, GPT-4o supports retrieval via the logprobs parameter, enabling direct application of our method. For APIs that do not expose logits, attackers would need to simulate the generation process and approximate logits using multiple sampling. This requires storing more than just greedy completions—attackers would need to cache the top few candidate tokens at each step and re-run completions accordingly, leading to exponential growth in sampling cost. Despite the overhead, we note that even partial completions (e.g., a few-token prefix) may still yield meaningful extraction.
>
> Overall, we agree that API-accessible but closed-source scenarios introduce additional complexity. Investigating how to perform extraction effectively under such constraints is an important direction for future work.
>
> 3. The attack performance on chatmodels:
>
> >It’s unclear whether the attack also performs well on chat models.
>
> >Can the authors please elaborate on how their method would work on chat models? I'm not really expecting for additional experimental results considering that will incur a lot of load, just sharing your thoughts will suffice.
>
> Thank you for the question. We did not include experiments on chat models primarily because obtaining an exact unlearned version of a chat model is non-trivial. Chat models are typically derived from base models via additional fine-tuning stages, such as instruction tuning. To simulate unlearning in this context, one would first need to redo the base fine-tuning on a dataset that excludes the unlearning set and then reapply the instruction-tuning process. This procedure is both computationally expensive and complex to implement.
>
> However, we hypothesize that data extraction on chat models would require more carefully designed prefixes to elicit training data, as there is often no fixed prompt structure that directly points to the data. Prior work (e.g., [2]) has shown that specially designed attacks can make chat models even more vulnerable, suggesting that crafting the right instructions could amplify extraction effectiveness.
>
> Nonetheless, since our approach is based on contrasting the model’s behavior before and after unlearning, the core mechanism remains applicable. As long as both models are given the same input prompt, we expect our guidance-based method to still provide a significant boost in extraction performance compared to unguided generation in the chat model setting.
>
> [2] Nasr, Milad, et al. "Scalable extraction of training data from (production) language models." arXiv preprint arXiv:2311.17035 (2023).
>
>
>
> 4. About availability of exact unlearning:
>
> > The attack assumes exact unlearning, which I’m not sure is commonly adopted in practice. A previous work points out that exact unlearning is “too costly to be practical”
>
> > Were there any real cases where exact unlearning was used on an LLM?
>
> We note that existing benchmarks such as MUSE and TOFU adopt exact unlearning as a reference point that reflects the ideal outcome of unlearning. Recent efforts have also focused on improving the efficiency and practicality of exact unlearning methods [3,4], as detailed in Sec. 2.1. We therefore summarize it as the de facto gold standard, and our main focus is to examine whether privacy risks still remain under this setting.
>
> As an extension, we also experiment under approximate unlearning methods in Appendix Sec. C, where our extraction method consistently outperforms the baseline. The effectiveness of extraction improves when the approximate unlearning better preserves model utility.
>
> [3] X. Xia, Z. Wang, R. Sun, B. Liu, I. Khalil, and M. Xue. Edge unlearning is not" on edge"! an adaptive exact unlearning system on resource-constrained devices. arXiv preprint arXiv:2410.10128, 2024.
>
> [4] K. Kuo, A. Setlur, K. Srinivas, A. Raghunathan, and V. Smith. Exact unlearning of finetuning data via model merging at scale. arXiv preprint arXiv:2504.04626, 2025.

---

> > ### Comment · Reviewer_SsS4 · 2025-08-03
> > **Response to rebuttal**
> >
> > Thank you for the rebuttal.
> >
> > > For the API-only setting, wouldn’t the attacker need access to both the pre-unlearned and post-unlearned API during inference time, unlike how the authors claim in Line 116?
> >
> > Can you please clarify if I misunderstood this part?

---

> > > ### Author Response · Authors · 2025-08-04
> > > **Official Comment by Authors**
> > >
> > > Thank you for your comments. If both the pre- and post-unlearning logits APIs are accessible, then our attack applies directly. However, our statement in Line 116 targets a broader class of API-based deployments, where the attacker has interacted with the logits API prior to unlearning and **cached logits** for future use (before unlearning happens). Specifically, the attacker can prepare by issuing multiple queries and generating multiple continuations. At each decoding step, the attacker **saves the logits associated with these alternative generation paths**, rather than just the logits of the greedy output.
> > >
> > > These cached logits can later be compared against the logits from the post-unlearning model to compute guidance signals, which are then used to steer generation toward trajectories that may still reflect the forgotten data distribution. While the number of such paths grows exponentially with the number of positions and branches explored, we note that in practice, storing only a subset—such as a few top‑k continuations—and modifying only the first few positions could already offer some advantage over unguided generation. We will clarify this scenario and include the corresponding discussion in the revised version.

---

> > > > ### Comment · Reviewer_SsS4 · 2025-08-05
> > > > **Response to author comment**
> > > >
> > > > Thank you for your clarification.
> > > >
> > > > I do not have further discussion points.

---

### Decision · Program_Chairs · 2025-09-17

**Decision:**

Accept (poster)

**Comment:**

The paper proposes a data extraction attack that on LLMs that have been unlearned using exact unlearning. Therefore, it relies on signals from the model before and after the unlearning step, which corresponds, for example to having access to previous checkpoints. In particular, it uses the before-unlearning modal to guide the generation and also compares the token probabilities before and after unlearning, filtering additionally by the high-probability tokens in the model before unlearning.

**Strengths**: The reviewers appreciate the paper's clear presentation and thorough experimentation on multiple benchmarks and with two models (and a third one, Mixtral-8x7B being added during the rebuttal). They highlight that the findings are novel and interesting, revealing a previously undiscovered privacy leakage under exact unlearning, given access to different checkpoints of the model.

**Weaknesses**: Initial concerns were raised regarding the question whether the threat model is realistic. There were successfully addressed during the rebuttal, highlighting the importance in Open Weights models and in APIs that expose different model checkpoints over time. Another concern remains about the dependence of the method on a good set of hyperparameters, however, given that the method outperforms baselines under various values for $w$, the practical relevance of the concern might be limited.

Overall, the reviewers found the framing of the paper too strong and would like to encourage the authors to tune down tone according to the rebuttal: this includes changing the title and framing to make clear that it is the deployment of unlearning which is relevant, not that unlearning itself is inherently broken.